# Glycocalyx Acts as a Central Player in the Development of Tumor Microenvironment by Extracellular Vesicles for Angiogenesis and Metastasis

**DOI:** 10.3390/cancers14215415

**Published:** 2022-11-03

**Authors:** Ye Zeng, Yan Qiu, Wenli Jiang, Bingmei M. Fu

**Affiliations:** 1Institute of Biomedical Engineering, West China School of Basic Medical Sciences and Forensic Medicine, Sichuan University, Chengdu 610041, China; 2Department of Biomedical Engineering, The City College of the City University of New York, New York, NY 10031, USA

**Keywords:** extracellular vesicles, glycocalyx, microenvironment, angiogenesis, metastasis

## Abstract

**Simple Summary:**

The glycocalyx is a fluffy sugar coat covering the surface of all mammalian cells. While glycocalyx at endothelial cells is a barrier to tumor cell adhesion and transmigration, glycocalyx at tumor cells promotes tumor metastasis. Angiogenesis at primary tumors and the growth of tumor cells at metastatic sites are all affected by the tumor microenvironment, including the blood vasculature, extracellular matrix (ECM), and fibroblasts. Extracellular vesicles (EVs) secreted by the tumor cells and tumor-associated endothelial cells are also considered to be the components of the tumor microenvironment. They can modify tumor vasculature, ECM, and fibroblasts. But how the EVs are generated, secreted, and up taken by the endothelial and tumor cells in the development of the tumor microenvironment are unclear, especially after anti-angiogenic therapy (AAT). The objective of this short review is to summarize the role of the glycocalyx in EV biogenesis, secretion, and uptake, as well as the modulation of the glycocalyx by the EVs.

**Abstract:**

Angiogenesis in tumor growth and progression involves a series of complex changes in the tumor microenvironment. Extracellular vesicles (EVs) are important components of the tumor microenvironment, which can be classified as exosomes, apoptotic vesicles, and matrix vesicles according to their origins and properties. The EVs that share many common biological properties are important factors for the microenvironmental modification and play a vital role in tumor growth and progression. For example, vascular endothelial growth factor (VEGF) exosomes, which carry VEGF, participate in the tolerance of anti-angiogenic therapy (AAT). The glycocalyx is a mucopolysaccharide structure consisting of glycoproteins, proteoglycans, and glycosaminoglycans. Both endothelial and tumor cells have glycocalyx at their surfaces. Glycocalyx at both cells mediates the secretion and uptake of EVs. On the other hand, many components carried by EVs can modify the glycocalyx, which finally facilitates the development of the tumor microenvironment. In this short review, we first summarize the role of EVs in the development of the tumor microenvironment. Then we review how the glycocalyx is associated with the tumor microenvironment and how it is modulated by the EVs, and finally, we review the role of the glycocalyx in the synthesis, release, and uptake of EVs that affect tumor microenvironments. This review aims to provide a basis for the mechanistic study of AAT and new clues to address the challenges in AAT tolerance, tumor angiogenesis and metastasis.

## 1. Introduction

Malignant tumors remain a major threat to human health. Two key challenges facing conventional oncology therapies such as surgery, radiation, and chemotherapy are the local invasion of primary tumors and the metastasis of primary tumors to form secondary tumors, and the development of tumor cell tolerance to therapies [1]. Angiogenesis is a crucial process in tumor growth and progression. In fact, solid tumors do not grow beyond 2–3 mm^3^ without angiogenesis. Moreover, the vascular system has important functions such as maintaining nutrient and oxygen supply, waste removal, and immune surveillance. Therefore, anti-angiogenic therapies (AATs) that target tumor vasculature to inhibit the growth and metastasis of solid tumors have been widely used in the clinic [2]. However, AATs such as vascular pruning, disruption, and normalization have failed to achieve the desired therapeutic outcomes [1]. After the termination of AATs, the angiogenic process is reactivated and the tumor vascular system is reconstructed, leading to rapid tumor recurrence and metastasis. For example, tumors grow more rapidly in patients with metastatic colorectal cancer after the termination of bevacizumab treatment [3]. Surgical treatments may also increase plasma vascular endothelial growth factor (VEGF) levels and accelerate colorectal cancer growth and metastasis [3]. Discontinuous treatment with bevacizumab promoted tumor growth and revascularization in colorectal cancer xenograft mice [4]. Although sunitinib and bevacizumab reduced microvessel density in the primary tumor tissue of patients with renal cell carcinoma, however, sunitinib promoted endothelial cell (EC) proliferation in tumor tissues [5]. Angiogenesis in tumor growth and tumor progression involves a series of complex changes in the microenvironment. But the mechanism of AAT resistance remains unclear at present. In particular, the mechanisms by which AATs modulate the tumor microenvironment remain unclear.

Recently, we reviewed the following several mechanisms of AAT resistance in tumors [6,7]: (1) hypoxia due to tumor vascular pruning and increase in subsequent angiogenic factors such as VEGF, (2) microenvironments related to different phenotypic characteristics of ECs in tumor and in normal tissues and the evolution of endothelial heterogeneity, (3) extracellular vesicles (EVs)-mediated crosstalk between tumor cells and ECs, (4) inability of VEGF to be effectively recognized by therapeutic agents because of packaging and masking of VEGF by EVs, and (5) stromal stiffness of the tumor. Although it is currently well recognized that the tumor microenvironment is an important mechanism underlying AAT resistance, the lack of understanding of the formation and maturation of the tumor microenvironment has severely limited the development of novel AAT strategies.

The tumor microenvironment provides a suitable soil for tumor angiogenesis and growth. Tumor vasculature is disorganized and tortuous, with abnormal vascular ECs and leaky vascular walls. Tumor angiogenesis arises from the stimulation of a complex set of genetic and microenvironmental factors. Interstitial fluid pressure, hypoxia, and acidosis can regulate tumor angiogenesis [8]. EVs are important components of the tumor microenvironment and share many common biological properties, although they can be classified into different types, such as exosomes, apoptotic vesicles, and matrix vesicles, according to their origin and characteristics. EVs perform their biological functions by carrying and transporting ribonucleic acids, proteins, and lipids [9]. EVs are important targets of microenvironmental modification and play an important role in tumor angiogenesis [10]. Compared to normal vascular ECs, tumor ECs are heterogeneous, and AAT can induce tumor ECs to secrete exosomes carrying VEGF, which is defined as VEGF exosomes that could in turn induce angiogenesis and the progression of hepatocellular carcinoma (HCC) [11].

Vascular ECs and their luminal side of the endothelial glycocalyx layer (EGL) are the basic barrier and regulator of material exchange between circulating blood and tissues [12]. Previous studies have shown that the intercellular space between vascular ECs is an important pathway for material exchange in the microcirculation and an important site for tumor cell adhesion and extravasation [13,14,15,16,17]. The EGL also acts as a natural barrier for tumor cells to metastasize across the endothelium [13,14,15,16] and mediates the secretion and uptake of EVs [18,19]. AATs triggered tumor endothelial VEGF exosomes, which are potential modifiers of the tumor stromal microenvironment, are an important linkage between vascular ECs and tumor cells [11]. On the other hand, many components carried by EVs can modify the EGL, which finally facilitates the development of the tumor microenvironment. To better understand the role of the glycocalyx in AATs, and in developing the microenvironment for angiogenesis and metastasis, we hereby review the role of the glycocalyx in exosome synthesis and release, and the modification or modulation of the glycocalyx by the cargo of EVs. The changes of glycocalyx by VEGF exosomes may promote tumor angiogenesis and tumor cell metastasis. The glycocalyx and its roles in exosomal VEGF synthesis, release, and uptake may help us understand AAT resistance from a new perspective.

## 2. Development of Tumor Microenvironment by EVs

The tumor microenvironment is an interactive cellular environment surrounding the tumor whose main function is to establish cellular communication pathways supporting tumorigenesis [20]. EVs secreted by tumors have been well documented to aid cancer progression and invasion by modulating the tumor microenvironment. Figure 1 depicts the schematic for how EVs contribute to the development of the tumor microenvironment. Table 1 summarizes the EV-regulated tumor microenvironment in different types of cancer.

### 2.1. Cancer-Associated Fibroblasts and Extracellular Matrix (ECM) Remodeling

The EVs derived from tumor cells contribute to the formation of cancer-associated fibroblasts, inducing the remodeling of tumor stroma via transfer of pro-metastatic factors such as non-coding oncogenic micro ribonucleic acids (miRNAs) [37]. EV-mediated transferring of miR-200 from metastatic colorectal cancer cells to recipient fibroblasts promotes myofibroblast differentiation and fibronectin expression, shaping the stromal landscape, which is likely to be independent of local TGF-β availability [21]. EVs from SW620 colorectal cancer cells activate fibroblasts to invade the ECM through upregulation of pro-invasive regulators of membrane protrusion (PDLIM1, MYO1B) and matrix-remodeling proteins (MMP11, EMMPRIN, ADAM10) [22]. Melanoma-derived exosomes reprogram fibroblasts into tumor-promoting cancer-associated fibroblasts by delivering Gm26809 and promoting the proliferation and migration of melanoma cells through ECM remodeling [23]. Membrane type 1 matrix metalloproteinase (MT1-MMP, MMP14) is a highly efficient ECM degrading enzyme that plays an important role in tissue homeostasis and cell invasion. It has been found that the exosomes secreted by melanoma cells contain MT1-MMP, which can degrade collagen type 1 and gelatin, leading to ECM remodeling [24].

### 2.2. Immune Landscape in Tumor Microenviornment

In turn, tumor cells utilize EVs to evade the immune surveillance by inducing apoptosis in immune cells through the transfer of immunosuppressive molecules such as HSP72 [31], and Fas-L [32]. Exosomal lncRNA KCNQ1OT1 derived from colorectal cancer cells promotes colorectal cancer immune escape through miR-30a-5p/USP22 regulated PD-L1 ubiquitination [33]. Cervical cancer-secreted exosomal miR-1468-5p epigenetically activates the JAK2/STAT3 pathway in lymphatic endothelial cells by directly targeting homeobox containing 1 (HMBOX1) in the SOCS1 promoter, activating an immunosuppressive program that allows cancer cells to escape anti-cancer immunity [34]. Exosomes isolated from the serum of nasopharyngeal carcinoma (NPC) patients or from the supernatant of TW03 cells impair T-cell function by inhibiting T-cell proliferation and Th1 and Th17 differentiation and promoting Treg induction by NPC cells in vitro [35]. In cervical squamous cell carcinoma (CSCC), miR-142-5p is transferred by CSCC-secreted exosomes into lymphatic endothelial cells to exhaust CD8+ T cells via suppressing lymphoid-rich interaction domain protein 2 (ARID2) to upregulate lymphatic indoleamine 2, 3-dioxygenase expression [36].

### 2.3. Tumor Neovascularization

The EVs secreted by tumor cells within the tumor microenvironment transfer angiogenic factors such as EGFR and VEGF into ECs, inducing a pro-angiogenic niche to support the pathological angiogenesis and tumor growth [38,39]. The role of VEGF in the tumor development and progression is complicated. VEGF activates the resting ECs and tip ECs to direct neovascularization toward a hypoxic, nutrient-deficient microenvironment. In vitro, it works in a low serum concentration of 0.1% to promote EC proliferation and migration. However, the tumor microenvironment is characterized by high serum concentrations, and the ascites proteins in tumor patients can be up to 25 g/dL [40]. VEGF may not exert biological effects on the proliferation and migration of ECs as one would expect during tumor development in vivo. Recent studies have shown that tumor growth in patients with glioma is not associated with the VEGF pathway [41]. Although high-dose bevacizumab (Avastin) specifically binds to VEGF, its inhibitory effect on gliomas is independent of angiogenesis. The increased angiogenesis and permeability establish a pre-metastatic niche where the primary tumor can promote its own metastasis by recruiting stromal cells to distant organs or modulating the gene expression of distant cells, which is partially mediated by EVs [42]. Moreover, it is suggested that VEGF exosomes are closely associated with the AAT resistance [11]. VEGF is also closely associated with tumor cell adhesion, immunosuppression [43,44], as well as vascular leakage [45]. VEGF can promote the adhesion of tumor cells to the microvascular endothelium [46]. VEGF is an important vascular permeability factor that may promote fibrin deposition and activation of tumor stroma cells by increasing vascular permeability [47].

The EVs can promote endothelial cell invasion into tissues and accelerate neovascularization, which is closely associated with the secretion of matrix metalloproteinases (MMPs) [48]. In turn, MMPs regulate the release of EVs. For example, EVs containing MT1-MMP facilitate the matrix degradation to promote tumor invasion and metastasis [49]. Knockout of MMP3 reduces tumoroid (tumor organoid) size and development of the necrotic area within tumoroids via releasing broken EVs from tumoroids [50]. MMP13 is overexpressed in nasopharyngeal carcinoma cells and carried by exosomes, contributing to the tumor cell metastasis and angiogenesis [26].

Moreover, tumor-derived exosomes contain various miRNAs that participate in the tumor and stromal cell interaction, thus contributing to the tissue remodeling of the tumor microenvironment. Under hypoxia, miR-23a was significantly upregulated in lung cancer exosomes, which increases angiogenesis and vascular permeability by targeting prolyl hydroxylase and the tight junction protein ZO-1 [25]. MiR-21-5p was upregulated in exosomes from papillary thyroid cancer BCPAP cells under hypoxic conditions, which increased endothelial tube formation via suppressing TGFBI and COL4A1 [27]. Exosomal miR-183-5p promotes angiogenesis in colorectal cancer by regulation of FOXO1 [28]. Exosomal miR-25-3p transferred from colorectal cancer cells to ECs increases vascular permeability and angiogenesis via targeting KLF2 and KLF4 [29]. Delivery of miR-21-5p from colorectal cancer cells to ECs by exosomes promotes angiogenesis and vascular permeability in colorectal cancer via targeting Krev interacting capture protein 1 (KRIT1) on recipient ECs [30]. VEGF was also increased by exosomal miR-25-3p and miR-21-5p.

## 3. The Relationship between Glycocalyx and Tumor Microenvironment

The intercellular cleft between vascular ECs is a major pathway for water and hydrophilic molecule exchange in microcirculation and a major site for tumor cell adhesion and transmigration. The endothelial glycocalyx layer is located at the vascular luminal surface, in direct contact with blood, covers the intercellular cleft, and is a natural barrier for tumor cells to metastasize across the vascular wall or endothelium. EGL has a complex structure and is comprised of many components. Heparan sulfate proteoglycan (HSPGs), syndecan-1 (SDC1), syndecan-2 (SDC2), syndecan-4 (SDC4), and glypican-1 (GPC1), and their glycosaminoglycan side chains, including heparan sulfate (HS), hyaluronic acid (HA), and chondroitin sulfate (CS), are important components of EGL [51].

Both vascular endothelial and tumor cells in the tumor microenvironment are involved in tumorigenesis and metastasis. The microenvironmental modification and glycocalyx remodeling of vascular endothelium and tumor cells are prerequisites for tumor angiogenesis, tumor cell invasion, and metastasis. Tumor cell adhesion to and transmigration across the vessel wall are accompanied by the degradation of endothelial glycocalyx and disruption of EC junctions [13,14,15,16]. At the branches and turns of microvessels, the endothelial glycocalyx is more likely to be destroyed by flow-induced factors, which increase vascular permeability and make it easier for tumor cells to adhere to the exposed adhesion molecules either on the ECs or in the ECM [13,14,15,16,17]. In contrast, tumor cell glycocalyx is closely associated with its ability of migration [52]. A most recent study found that a tumor secretion, VEGF, while disrupting the glycocalyx of human cerebral microvascular endothelial cells, significantly enhances heparan sulfate and hyaluronic acid coverage on malignant breast cancer cells MDA-MB-231 [53]. Tumor cell surface glycocalyx promotes the uptake and internalization of EVs [18]. The secretion of syntenin exosomes also requires glycocalyx [54,55]. Disruption of the tumor cell surface glycocalyx reduces the metastatic tumor cells by 95% [52]. These findings suggest that glycocalyx is a potential mediator for vascular endothelium-tumor cell interaction. It is thus important to elucidate the regulatory role of cell surface glycocalyx in tumor microenvironment modulation. Table 2 summarizes the association of glycocalyx components with the tumor microenvironment in various cancers. Before specifying each role of the glycocalyx, we first review the modification of the glycocalyx by the EVs with various cargos.

### 3.1. Modification of Glycocalyx by EVs

The glycocalyx on different cells may have different changes in response to external stimuli. Vascular ECs co-cultured with tumor cells secrete EVs carrying vascular endothelial cadherin (VE-cadherin), which are reused by tumor cells and promote tumor angiogenesis [10]. VE-cadherin is closely linked to the vascular endothelial glycocalyx. Degradation of the vascular endothelial glycocalyx inhibits circumferential strain-induced VE-cadherin transcription [56], while VE-cadherin knockdown can result in a reduction of the vascular endothelial glycocalyx [57]. Plasma-derived EVs secreted by linoleic acid-induced tumor cells or EVs from the serum of cancer patients promote MMPs secretion and angiogenesis [58]. Previous studies have shown that MMPs can degrade the vascular endothelial glycocalyx [59]. These findings indicate that the glycocalyx can be modified by EVs.

**Table 2 cancers-14-05415-t002:** Components of glycocalyx and its association with tumor microenvironment.

Cancer Type	Components	Pathway	Function	Ref.
Multiple myeloma	SDC1	HGF/c-met/IL-11	ECM remodeling	[60]
Breast cancer	SDC1	TGF-β	ECM remodeling	[61]
Breast cancer	SDC1	FGF2, SDF1	Growth signal	[62]
Bladder cancer	SDC4	NF-κB	ECM deposition of Tenascin-C	[63]
Pancreatic ductal adenocarcinoma	SDC1	CCL5	Mediate the T cells crosstalk with tumor cells	[64]
Breast cancer	SDC2	TGF-β	Immune evasion	[65]
Melanoma	SDC3		Proinflammatory response	[66]
Colon cancer	SDC1	VEGFR2	Angiogenesis	[67]
Colon cancer	SDC2	VEGF	Angiogenesis	[68]

### 3.2. Regulation of ECM Remodeling by Glycocalyx

Chemotherapy-driven shedded SDC1 stimulates IL-11 via enhancing HGF/c-met signaling, and it might bind to bone marrow ECM molecules such as collagen and fibronectin, worsening the bone disease in myeloma [60]. SDC1 overexpression in senescent breast stromal fibroblasts induced by ionizing radiation treatment of breast cancer via an autocrine action of TGF-β reduces expression of COL1A1 and increases expression of several MMPs, i.e., MMP-1, -2, -3, and -9 [61]. T47D breast carcinoma cells induce expression of SDC1 in mammary fibroblasts, whereas shedding of SDC1 HS from the fibroblast surface mediates the paracrine growth signal of breast carcinomas [62]. An ECM component, Tenascin-C, which has been reported to compete with the binding sites of fibronectin with SDC4, is recently reported to activate NF-κB signaling by binding with SDC4 to promote tumor progression [63].

### 3.3. Regulation of Immune Landscape by Glycocalyx

A recent single-cell RNA-seq analysis demonstrated that CCL5-SDC1/4 receptor-ligand interaction mediates the T cells’ crosstalk with tumor cells in pancreatic ductal adenocarcinoma [64]. SDC2 enhances TGF-β signaling in tumor-associated stromal cells and mediates immune evasion in breast cancer [65]. SDC3 expressed on tumor-associated macrophages is promoted by hypoxia inducible factors (HIFs) and might link to a proinflammatory response [66].

### 3.4. Regulation of Angiogenesis by Glycocalyx

It has been shown that SDC1 promotes the transformation of tumor ECs into an angiogenic phenotype [67]. SDC1 silencing inhibits the organization of tumor ECs into patent vessels in severe combined immunodeficient (SCID) mice, reduces membrane expression of VEGFR2, and thus weakens the colocalization of VEGFR2 with SDC1 [67]. SDC2 is closely associated with cytoskeleton organization, integrin signaling, and developmental angiogenesis, and is required for the development of the vascular system. SDC2 promotes VEGF/VFGFR2 complex formation and VEGF-dependent neovascularization [69]. Global and inducible endothelial-specific deletion of SDC2 in mice markedly impairs VEGF signaling and leads to angiogenic defects [69]. The shed SDC2 enhances tumorigenic activity by increasing the crosstalk of cancer cells with tumor-associated macrophages and endothelial cells to enhance angiogenesis for colon cancer progression via producing VEGF [68]. VEGF is a central effector of angiogenesis and vascular permeability regulated by EVs. VEGF also activates MMPs [70], which can degrade the vascular endothelial glycocalyx [59]. The interplay among VEGF, MMPs, and endothelial glycocalyx is still unclear.

Moreover, whether vascular endothelial glycocalyx participates in the tumor angiogenesis and tumor metastasis remains unclear. Further investigation is required. VEGF not only selectively regulates the proliferation and motility of vascular ECs but also enhances vascular permeability, leading to fibrin gel deposition and providing a suitable microenvironment for tumor metastasis [47]. VEGF can disrupt EC junctions, promote the adhesion of tumor cells to microvessels, and enhance microvascular permeability through the VEGF receptor 2 (VEGFR2/FDR/Flk-1) pathway [16,46]. The role of the VEGFR2 pathway in mediating the disruption of vascular endothelial glycocalyx and cell junctions has attracted much attention recently.

## 4. Regulation of Synthesis, Release, and Uptake of EVs by Glycocalyx

For intracellular communication, tumor cells capture and endocytose the EVs. This process requires the glycocalyx [18,19]. Currently, researchers have focused on identifying exosome subtypes and exploring the biological functions of exosomes and the underlying mechanisms through the cargo of exosomes [71]. SDC1 can promote the formation of specific tumor microenvironments by altering the packaging of miRNAs and circRNAs by exosomes [72]. One of the C-terminal fragments (CTFs) of SDC1 and SDC4, syntenin, is a marker of exosomes. The secretion of syntenin-containing exosomes is dependent on HS and SDC4 [54]. Heparanase can regulate the synthesis of SDC–syntenin exosomes [55]. Targeting the PDZ2 domain of syntenin using compounds can reduce exosomal syntenin and SDC4, but not other exosomal markers [73]. In contrast, tumor cell HSPGs can mediate exosome uptake [74]. SDC4 mediates the uptake and internalization of hydroxyapatite nanoparticles or exosomes in tumor cells, promoted by fetuin-A and histone [19]. HSPGs can interact with ITGB3 to capture exosomes and accelerate endocytosis-mediated exosome internalization [18]. Figure 2 demonstrates the schematic for the potential roles of glycocalyx in the biogenesis, uptake, and internalization of EVs, while Table 3 summarizes the glycocalyx components and their associations with exosomes or EVs.

### 4.1. Synthesis and Secretion of Exosomes by Glycocalyx

The stimulatory effect that heparanase has on exosome biogenesis is related to its ability to promote intraluminal budding and the formation of exosomes by enzymatically altering the heparan sulfate chain of syndecans [39,73]. Transfection with cDNA for heparanase in ARH-77 human lymphoblastoid cells increased exosome secretions [39]. The addition of exogenous recombinant heparanase to MDA-MB-231 breast cancer cells also induces exosome secretion. Exosomes secreted by osteoblasts could inhibit the differentiation of osteoclast progenitor cells via miR-503-3p/Hpse (heparanase gene), thus regulating the balance of bone formation and absorption [78]. Moreover, the secretion of syntenin-containing exosomes is dependent on HS and SDC4 [54]. Mechanistically, heparanase enhances exosome biogenesis by stimulating the endocytosis of syndecans and shortening the heparan sulfate chains of syndecans through the syntenin-Alix pathway [55]. Heparanase promotes exosome secretion from glioma cells, and those exosomes could partially restore the sensitivity of U251 cells to temozolomide [75]. The has_circ_0042003 gene was upregulated in those exosomes, which confer the resistance of U251 cells to temozolomide [75].

### 4.2. Uptake and Internalization of Exosomes by Glycocalyx

Heparanases that are enhanced in human cancer cells such as myeloma, lymphoblastoid, and breast cancer, induces exosome secretion and increases levels of protein cargo in exosomes, which stimulate the migration of tumor cells on fibronectin and the invasion of endothelial cells through the ECM [39]. For example, the treatment of bortezomib increased heparanase expression and heparanase-loaded exosomes in myeloma patients [76]. The heparanase can be detected by flow cytometry, and it is thus believed to be present on the exosome surface, where it is capable of degrading heparan sulfate on SDC1, embedded within the ECM. Monoclonal antibody H1023 can inhibit the migration of macrophages and the secretion of myeloma growth factor TNF-α induced by those exosomes [76], confirming the role of heparanase in the modification of the ECM via exosome secretion.

The degradation of heparan sulfate contributes to the remodeling of the ECM that alters tumor and host cell behaviors, which likely contributes to chemoresistance and eventual relapse [76]. Additionally, evidence has demonstrated that tumor cell glycocalyx promotes the uptake and internalization of EVs [18]. Tumor cell HSPGs mediate the uptake of exosomes [74]. SDC4 mediates the uptake and internalization of hydroxyapatite nanoparticles or exosomes in tumor cells, which are promoted by methemoglobin-A (fetuin-A) and histones [19]. HSPGs interact with ITGB3 to uptake EVs and accelerate endocytosis-mediated EV internalization [18].

Interestingly, higher levels of SDC1, VEGF, and hepatocyte growth factor (HGF) are present in exosomes secreted by heparanase high-expressing cells as compared with heparanase-low-expressing cells. The exosomes from heparanase-high-expressing cells could enhance the endothelial cell invasion [39]. Reinforced expression of heparanase or delivery of recombinant heparanase in myeloma tumor cells increases the gene expression of VEGF and HGF and promotes the shedding of SDC1 via upregulating MMP-9 [79,80,81]. It is speculated that heparan sulfate facilitates the capture of exosomes on recipient cells by serving as a key receptor for fibronectin or exosome cargo [77]. Likewise, the enzymatic depletion of cell-surface heparan sulfate proteoglycans or pharmacological inhibition of endogenous proteoglycan biosynthesis by xyloside significantly attenuates exosome uptake in cancer cells [74].

## 5. Summary and Perspective

Here, we review the role of the glycocalyx in EV synthesis, release, and uptake, and the modification or modulation of the glycocalyx by the cargo of EVs. The present evidence supports that syndecans, their heparan sulfate chains, and the modification enzyme heparanase all play critical roles in the secretion and uptake of EVs. The function of EVs, which is closely associated with their compositions, is regulated by the glycocalyx. Given the importance of exosomes in the microenvironment of diseases such as cancer and inflammation, the glycocalyx compositions that are responsible for exosome secretion and uptake are viable targets. As exosomes diffuse through the tumor microenvironment, the ability of exosomal heparanase to shed heparan sulfate could be critical for cell-ECM interaction during tumor metastasis and/or inflammation. Application of the heparanase inhibitor Roneparstat can enhance anticancer chemotherapy efficacy and overcome drug resistance [82] and also attenuate the inflammatory cytokine release from human macrophages through disruption of NF-κB signaling [83]. The heparin mimic Roneparstat is presumed to block exosome docking by competing against cell surface heparan sulfate. The monoclonal antibody that inhibits heparanase enzyme activity could suppress the exosomes-stimulated migration of macrophages and secretion of TNF-α [76]. In addition, the heparan sulfate chains on the exosome surface interact with fibronectin, modulating cell-ECM adhesion and cell migration in either positive or negative ways [74,77]. The roles of syndecans and heparanase in the regulation of the tumor microenvironment and in EV biogenesis and docking should be further studied to develop the therapeutic strategy targeting exosomes.

Particularly, our previous studies have shown that VEGF can disrupt endothelial intercellular junctions, promote the adhesion of tumor cells to microvessels, and enhance microvascular permeability through the VEGFR2 pathway [16,46]. It is possible that, at the primary site of the tumor, AAT induces tumor endothelial cells to secrete VEGF exosomes through the endothelial glycocalyx, and VEGF exosomes enhance tumor cell glycocalyx and tumor cell intravasation via vascular walls, causing a large number of tumor cells and VEGF exosomes to enter the circulation. On the other hand, at the metastatic site, circulating VEGF exosomes from the primary site bind to endothelial VEGFR2, exacerbating the degree of disruption of the endothelial glycocalyx and cell junctions, promoting vascular leakage, resulting in fibrin gel deposition, and providing a stromal microenvironment conducive to tumor cell adhesion and metastasis, eventually contributing to the tolerance of the AAT to tumors. The mechanistic study of AAT in biogenesis and the docking of VEGF exosomes should provide new clues to address the challenges of tumor metastasis and AAT resistance.

As previously reviewed, the heterogeneity of the EVs, the lack of isolation and purification standards, and the unknown optimal storage conditions and pharmacokinetics or biodistribution patterns are major obstacles to the development of exosomal agents [9]. There are proteins, lipids, and carbohydrates on the outer EV surface, which are difficult to decipher and are likely diverse across distinct EV subsets [84] and coordinated with the pericellular microenvironment. It will be of particular interest to find out the overall composition of exosomal cargos and, more specifically, the receptors and growth factors that interacted with glycocalyx or were dependent on glycocalyx for activity. The proteins on EVs have been widely explored by using proteomic analysis, which reveals a large amount of signaling pathways involved in health and disease [85]. However, to elucidate the role of the glycocalyx in the developing tumor microenvironment, the technical challenges that limit the analysis of lipids and glycocalyx on EVs must be overcome.

## 6. Conclusions

Tumor-derived EVs modulate the tumor microenvironment by inducing ECM remodeling, neovascularization, and immune escape. The glycocalyx is closely associated with the tumor microenvironment and plays critical roles in the synthesis, and packaging of the cargos that regulate crucial processes such as migration, invasion, and premetastatic niche formation, uptake, and internalization of EVs. Thus, glycocalyx acts as a central player in the development of the tumor microenvironment by EVs for tumor angiogenesis and metastasis.

## Figures and Tables

**Figure 1 cancers-14-05415-f001:**
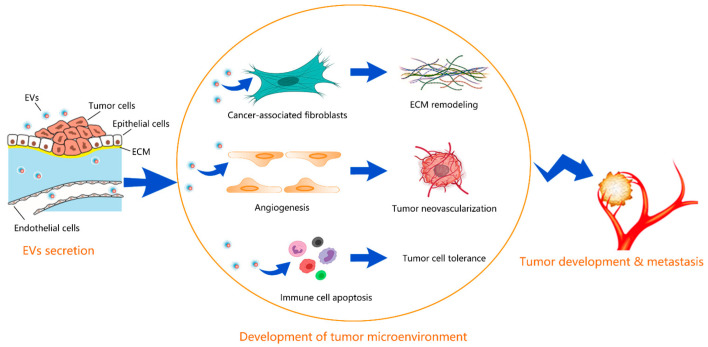
Schematic for how EVs contribute to the development of tumor microenvironment. EVs secreted from tumors promote the differentiation of fibroblasts into cancer-associated fibroblasts, inducing the ECM remodeling, promote the proliferation of endothelial cells and neovascularization, and help the tumor cells to evade the immune surveillance by inducing apoptosis in immune cells. The developed tumor microenvironment facilitates the tumor development and metastasis.

**Figure 2 cancers-14-05415-f002:**
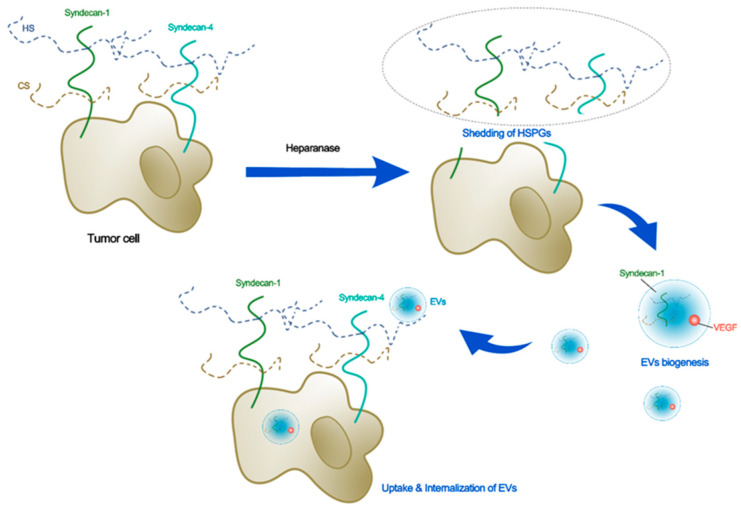
Schematic for the potential roles of glycocalyx in biogenesis, uptake, and internalization of EVs. Heparanase that sheds the tumor HSPGs induces secretion of EVs with higher levels of syndecan-1 and pro-angiogenic factors such as VEGF. The syndecan-4 plays an important role in secretion, uptake, and internalization of EVs in tumor cells.

**Table 1 cancers-14-05415-t001:** Tumor microenvironment regulated by EVs.

Cancer Types	EV Contents	Targets/Associated Pathways	Function	Ref.
Breast cancer	miR-200	ZEB1	ECM remodeling	[21]
Colorectal cancer		MMP11, EMMPRIN, ADAM10	ECM remodeling	[22]
Melanoma	lncRNA	Gm26809	ECM remodeling	[23]
Melanoma	MT1-MMP		ECM remodeling	[24]
HCC	VEGF from tumor ECs		AAT resistance of tumor ECs	[11]
Lung cancer	miR-23a	PHD1/2, ZO-1	Angiogenesis	[25]
Nasopharyngeal cancer	MMP-13		Angiogenesis	[26]
Thyroid cancer	miR-21-5p	TGFBI, COL4A1	Angiogenesis	[27]
Colorectal cancer	miR-183-5p	FOXO1	Angiogenesis	[28]
Colorectal cancer	miR-25-3p	KLF2, KLF4/VEGF	Angiogenesis	[29]
Colorectal cancer	miR-21-5P	KRIT1/VEGF	Angiogenesis	[30]
Colon carcinoma, lymphoma, and mammary adenocarcinoma	HSP72	STAT3 in myeloid-derived suppressor cells	Restrain immune surveillance	[31]
Melanoma	Fas-L		Lymphocyte apoptosis	[32]
Colorectal cancer	LncRNA KCNQ1OT1	miRNA-30a-5p/ USP22/PD-L1	Immune escape	[33]
Cervical cancer	miR-1468-5p	JAK2/STAT3 in lymphatic ECs	Immune escape	[34]
Nasopharyngeal carcinoma	miRNAs	MARK1	Impaired T-cell function	[35]
Cervical cancer	miR-142-5p	ARID2 in lymphatic ECs	Immunosuppressive	[36]

**Table 3 cancers-14-05415-t003:** Components of glycocalyx and their associations with exosomes/EVs.

Components/ Enzyme	Influence on EVs	Outcome to Tumor Microenvironment	Ref.
SDC1	miRNA packaged in lung cancer cells	Shapes the tumor microenvironment	[72]
Heparanase (Enzyme that sheds HS and SDCs)	Increased exosomes secretion in human lymphoblastoid cells and breast cancer cells	Tumor microenvironment bathed in much higher levels of exosomes	[39]
SDC4 and HS	Production of exosomes	Receptor trafficking	[54]
Heparanase	Biogenesis of exosomes	Specific cargo is probably selected through the interaction with HS and heparanase-trimmed HS	[55]
Heparanase	Exosomes secretion from glioma cells	Restores the sensitivity of glioma cells to temozolomide	[75]
Heparanase	Secretion of exosomes	Modification of ECM	[76]
HSPGs	Internalization of cancer cell exosomes	Glioblastoma multiforme (GBM) cell migration and signaling activation	[74]
SDC4	Uptake and internalization of hydroxyapatite nanoparticles or exosomes in tumor cells	Mediates cell motility and invasion of prostate cancer tumor cells	[19]
HSPGs	Uptake and internalization of exosomes in breast cancer cells	Interaction and activation of focal adhesion kinase by ITGB3 require these EVs	[18]
HS	Exosomes captured by myeloma cells	Serves as a key receptor for fibronectin or exosome cargos	[77]

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
