# Peer review of "Glycocalyx Acts as a Central Player in the Development of Tumor Microenvironment by Extracellular Vesicles for Angiogenesis and Metastasis"

_cancers, 2022, doi:10.3390/cancers14215415_

Round 1

Reviewer 1 Report

The short review by Zeng and Fu aims at summarizing the role of  glycocalyx and the generation of EVs and its effect on the tumor microenvironment. 

The language proficiency is acceptable through out with changes needed at some places.

1. Lines 105, 106 Sentences needs correction

2. Figure 1: Spelling check for Epithelial cells 

3. Lines 143, 273, 290 - Check sentence

4. Lines 170 -172 : Repetition - Same as lines 150-152.

5. Line 191: Needs modification

6. Line 249: Check spelling

7. It would add greatly to the review to add a section based on the type of cancer and  glycocalyx/EVs. And present that as a table for better understanding. 

Author Response

Reviewer 1
The short review by Zeng and Fu aims at summarizing the role of glycocalyx and the generation of EVs and its effect on the tumor microenvironment.
The language proficiency is acceptable through out with changes needed at some places.
1. Lines 105, 106 Sentences needs correction
2. Figure 1: Spelling check for Epithelial cells
3. Lines 143, 273, 290 - Check sentence
4. Lines 170 -172: Repetition - Same as lines 150-152.
5. Line 191: Needs modification
6. Line 249: Check spelling
Response to 1~6: Thank you for the careful review. The revisions were made accordingly. The major revisions are highlighted in the manuscript.
7. It would add greatly to the review to add a section based on the type of cancer and glycocalyx/EVs. And present that as a table for better understanding.
Response: Thanks for your suggestion. Three tables have been added. Please see Tables 1, 2 and 3.

Reviewer 2 Report

Major comments:

The authors reviewed the role of Glycocalyx as a central player in the development of tumor microenvironment by extracellular vesicles. Although they summarized some aspects of EV and molecules involved in cell-to-cell communications in tumor metastasis and anti-angiogenesis resistance, the review in its current form is far from a comprehensive review due to the following aspects.

The authors focused too much on the popular biomolecules like VEGF etc., involved in biomolecule metastasis and anti-angiogenesis resistance. Still, the real connection with the direct role of Glycocalyx and protein-copartners is largely lacking. Only sections 3 and 4 address thinly the aspect of Glycocalyx in EVs. Therefore it is far from a comprehensive review.

Authors should present Glycocalyx as a central player and discuss all the connections around these molecules.

Also, the review does not discuss the technical limitations in the lipids or carbohydrates best EV biomolecules. Several reviews have already highlighted that the EV field lacks correct technology, and many studies can be limited in their conclusions. For example, proteins are the significant EV-players. A review like DOI: 10.3390/proteomes7020022 and, for example, related to proteomics and including some from RNA (DOI: 10.1038/s41580-022-00460-3), lipid and sugars will help the reader to understand state of the art in the EV research. The most significant shortcoming in EV field is that there is not a universal and uniform technology worldwide. Hence, it makes it difficult for the readers to follow.  

This review is poorly constructed, excessively diffusive, and does not provide helpful insights. Publication in the current form is thus not recommended.

Minor comments:

References are lacking in most of the locations. For example, lines 37till and 52, several works of others are mentioned, but no references are included.

Similarly, in lines 59-97, in several places, no reference is cited—redundancy in text, repeating bits and pieces of the text in several paragraphs. 

For example, paragraph line 60 till 70 and paragraph line 71 till 83.

Author Response

Reviewer 2
Major comments:
The authors reviewed the role of Glycocalyx as a central player in the development of tumor microenvironment by extracellular vesicles. Although they summarized some aspects of EV and molecules involved in cell-to-cell communications in tumor metastasis and anti-angiogenesis resistance, the review in its current form is far from a comprehensive review due to the following aspects.
The authors focused too much on the popular biomolecules like VEGF etc., involved in biomolecule metastasis and anti-angiogenesis resistance. Still, the real connection with the direct role of Glycocalyx and protein-copartners is largely lacking. Only sections 3 and 4 address thinly the aspect of Glycocalyx in EVs. Therefore it is far from a comprehensive review. Authors should present Glycocalyx as a central player and discuss all the connections around these molecules.
Response: Thanks for your comments. We agree that this review is a short one. Only the direct role of glycocalyx and the protein-copartners are summarized to present glycocalyx as a central player.
Also, the review does not discuss the technical limitations in the lipids or carbohydrates best EV biomolecules. Several reviews have already highlighted that the EV field lacks correct technology, and many studies can be limited in their conclusions. For example, proteins are the significant EV-players. A review like DOI: 10.3390/proteomes7020022 and, for example, related to proteomics and including some from RNA (DOI: 10.1038/s41580-022-00460-3), lipid and sugars will help the reader to understand state of the art in the EV research. The most significant shortcoming in EV field is that there is not a universal and uniform technology worldwide. Hence, it makes it difficult for the readers to follow.
Response: Thanks for your comments. We have a recent review focusing on the features and challenges of exosomes (Ye Zeng et al., Front Cell Dev Biol. 2022 Jun 24;10:816698.), which includes the technical limitations. In the current review, we briefly described it in accordance with your suggestions. Please see below and Line 401-412 (highlighted).

“As previously reviewed, the heterogeneity of the EVs, the lack of isolation and purification standards, and the unknown optimal storage conditions and the pharmaco-kinetics or biodistribution patterns are major obstacles to the development of exosomal agents [9]. There are proteins, lipids, and carbohydrates on outer EV surface, which are difficult to decipher, and are likely diverse across distinct EV subsets [84] and coordinated with the pericellular microenvironment. It will be of particular interest to find out the overall composition of exosomal cargos and more specifically the receptors and growth factors interacted with glycocalyx or dependent on glycocalyx for activity. The proteins on EVs have been widely explored by using proteomic analysis, which reveals a large amount of signaling pathways involved in health and diseases [85]. However, to elucidate the role of glycocalyx in developing tumor microenvironment, the technique challenges that limit the analysis of lipids and glycocalyx on EVs have to be overcome.”
This review is poorly constructed, excessively diffusive, and does not provide helpful insights. Publication in the current form is thus not recommended.
Response: Thanks for your comments. We have reorganized the review, added 3 Tables as well as subsections for better understanding.

Minor comments:
References are lacking in most of the locations. For example, lines 37till and 52, several works of others are mentioned, but no references are included.
Similarly, in lines 59-97, in several places, no reference is cited—redundancy in text, repeating bits and pieces of the text in several paragraphs. For example, paragraph line 60 till 70 and paragraph line 71 till 83.
Response: Thanks for your comments. We have removed the redundant contents and cited the relevant references.

Reviewer 3 Report

I think the topic of “How glycocalyx influences tumor microenvironment via extracellular vesicles” is very interestingly and timely. However, I found that the current review needs significant restructuring/rewriting before publication because the content does not accurately match the title or subtitles.

(1)   The review is about how glycocalyx shapes the tumor microenvironment via exosomes. However, I noticed that the authors allocated a significant amount of space to discuss about the role of VEGF, VEGF-containing exosomes and angiogenesis during tumor development/progression. Perhaps the angiogenesis is his/her area of expertise. However, a review should give a fair overview of the field. I would recommend the authors either change their title/content (a review for how glycocalyx influence angiogenesis via exosome) or add more “meat” to the review. For example, the authors can discuss more on other important area of tumor biology…such as how glycocalyx influences exosome uptake/release in cancer-associated fibroblasts or how glycocalyx-regulated exosome synthesis/uptake shapes the immune landscape of the tumor microenvironment.

(2)   The authors sometimes added paragraphs/discussion that does not support the subtitles. For example, the Section 3 (Modification of glycocalyx by EVs) should discuss evidence of how EV modified the glycocalyx. However, they allocated a huge paragraph on the role of glycocalyx in tumor cancer cell migration/invasion and how glycocalyx could regulate tumor-EC interactions.

(3)   I like the section of how glycocalyx influences the exosome uptake/secretion. It will be useful for the readers if they can create a table to summarize type of proteoglycans, influence in exosome uptake/synthesis, and outcome to the tumor microenvironments.

Author Response

Reviewer 3
I think the topic of “How glycocalyx influences tumor microenvironment via extracellular vesicles” is very interestingly and timely. However, I found that the current review needs significant restructuring/rewriting before publication because the content does not accurately match the title or subtitles.
(1) The review is about how glycocalyx shapes the tumor microenvironment via exosomes. However, I noticed that the authors allocated a significant amount of space to discuss about the role of VEGF, VEGF-containing exosomes and angiogenesis during tumor development/progression. Perhaps the angiogenesis is his/her area of expertise. However, a review should give a fair overview of the field. I would recommend the authors either change their title/content (a review for how glycocalyx influence angiogenesis via exosome) or add more “meat” to the review. For example, the authors can discuss more on other important area of tumor biology…such as how glycocalyx influences exosome uptake/release in cancer-associated fibroblasts or how glycocalyx-regulated exosome synthesis/uptake shapes the immune landscape of the tumor microenvironment.
Response: Thanks for your comments. Accordingly, we have added new sections about cancer-associated fibroblasts and immune landscape of the tumor microenvironment. Please see sections 2.1, 2.2, 3.2 and 3.3, highlighted. We have also reorganized the review, added 3 Tables as well as subsections for better understanding.
(2) The authors sometimes added paragraphs/discussion that does not support the subtitles. For example, the Section 3 (Modification of glycocalyx by EVs) should discuss evidence of how EV modified the glycocalyx. However, they allocated a huge paragraph on the role of glycocalyx in tumor cancer cell migration/invasion and how glycocalyx could regulate tumor-EC interactions.
Response: Section 3 has been rewritten.
(3) I like the section of how glycocalyx influences the exosome uptake/secretion. It will be useful for the readers if they can create a table to summarize type of proteoglycans, influence in exosome uptake/synthesis, and outcome to the tumor microenvironments.
Response: Thanks for your suggestion. We have added Table 3 for this purpose.

Round 2

Reviewer 2 Report

All my comments have been addressed. 

Author Response

Thanks!

Reviewer 3 Report

The revised manuscript seems more structured at the latter subsections. However, the content still does not accurately match the title. The authors decided to keep the titles, but keep the abstract and introduction largely the same (still have included huge amount of space for angiogenesis information in the introduction/abstract section). An abstract and introduction should have an overall view of the whole review, this is not just about angiogenesis. Frankly, this review is still not suitable for publication at this stage and is very hard to read. The abstract and introduction are not a good overview of the review.

There are also obvious typos in the newly added sections (For example…2.1 Cancer-assocaited fibroblasts and ECM remdeling). The authors should pay more attention in proof reading before resubmission.

Author Response

The revised manuscript seems more structured at the latter subsections. However, the content still does not accurately match the title. The authors decided to keep the titles, but keep the abstract and introduction largely the same (still have included huge amount of space for angiogenesis information in the introduction/abstract section). An abstract and introduction should have an overall view of the whole review, this is not just about angiogenesis. Frankly, this review is still not suitable for publication at this stage and is very hard to read. The abstract and introduction are not a good overview of the review.

Reply: Thanks for your suggestion. We have modified the title to “Glycocalyx acts as a central player in the development of tumor microenvironment by extracellular vesicles for angiogenesis and metastasis”.

We also revised the abstract and added the following paragraph.

“In this short review, we first summarize the role of EVs in the development of the tumor microenvironment. Then we review how the glycocalyx is associated with the tumor microenvironment and how it is modulated by the EVs, and finally we review the role of glycocalyx in the synthesis, release, and uptake of EVs that affect tumor microenvironments.”

There are also obvious typos in the newly added sections (For example…2.1 Cancer-assocaited fibroblasts and ECM remdeling). The authors should pay more attention in proof reading before resubmission.

Reply: Thanks for your careful review. We have thoroughly checked the manuscript and corrected the typos.

Round 3

Reviewer 3 Report

The review flows better and content is more cohesive.